# Buckypaper-Based Nanostructured Sensor for Port Wine Analysis

**DOI:** 10.3390/s22249732

**Published:** 2022-12-12

**Authors:** Luiza Ferreira, Paula Pinheiro, Newton Barbosa Neto, Marcos Reis

**Affiliations:** 1Programa de Pós-Graduação em Ciência e Engenharia de Materiais, Universidade Federal do Pará, Ananindeua 67130-660, PA, Brazil; 2Programa de Pós-Graduação em Engenharia de Recursos Naturais da Amazônia, Instituto de Tecnologia, Universidade Federal do Pará, Belem 66075-110, PA, Brazil

**Keywords:** multiwalled carbon nanotubes, chemoresistive sensor, wine adulteration

## Abstract

The development of electronic gadgets has become of great relevance for the detection of fraud in beverages such as wine, due to the addition of adulterants that bring risks to human health as well as economic impacts. Thus, the present study aims to apply a buckypaper (BP) based on functionalized multiwalled carbon nanotubes (MWCNTs)/cellulose fibers as a sensor for the analysis of Port wine intentionally adulterated with 5 vol.% and 10 vol.% distilled water and ethyl alcohol. The morphology of BP characterized by scanning electron microscopy indicates the formation of agglomerates of random MWCNTs dispersed on the surface and between the fibers of the cellulosic paper. The analysis of the response of the film through the normalized relative resistance change showed a higher response of 0.75 ± 0.16 for adulteration with 10 vol.% of water and a mean response time of 10.0 ± 3.60 s and recovery of approximately 17.2 min for adulteration with 5 vol.% alcohol. Principal component analysis (PCA) was used in data processing to evaluate the ability of BP to recognize and discriminate analytes and adulterating agents, allowing the investigation of its potential application as a low-cost and easy-to-handle multisensor.

## 1. Introduction

The rigorous and objective assessment of the authenticity of food and beverages has become of paramount importance around the world, especially because of fraudulent activities involving adulteration. Adulteration is the term assigned to a food product that does not meet legal health and safety standards [1,2]. In the case of wine, its authenticity is protected by specific and strict regulations; however, due to its high economic value for some typical geographic areas of the world and associated sociocultural reasons, it is one of the beverages most subject to illegal practices [3,4].

False declaration of geographical origin, variety, year of production, dilution with water, the addition of exogenous substances (such as alcohol, sugars, preservatives and colorings) and mixtures with wines of inferior quality are common practices of adulteration in the wine industry and pose risks to human health. From a commercial point of view, the quality/price ratio can be illegally improved with an adulterated wine sold as high-quality wine, establishing unfair competition for regular production activities [3,4,5]. A study released in 2016 by the European Union Intellectual Property Office (EUIPO) on the economic impact of fraud in the spirits and wine sector estimated that legitimate industries lose approximately USD 1.41 billion in revenue annually in the European Union, resulting in direct losses of approximately 4800 jobs [6].

In recent years, modern analytical techniques such as mass spectrometry (MS), infrared or near-infrared spectroscopy (IR/NIR), and gas chromatography (GC) or high-performance liquid chromatography (HPLC) have been applied in wine analysis. However, despite presenting reliability and precision, these non-portable methods are relatively expensive and, in many cases, require sample preparation performed in specialized chemical laboratories and executed by qualified operators, thus arousing interest in the development of portable parametric systems that are easy to operate, have low cost and energy consumption and allow in situ measurements [2,7,8].

Multisensor systems for liquid phase analysis emerged as easy-to-handle and quick-response tools that can be applied in wine quality control. This allows qualitative analysis by classification according to organoleptic characteristics, presence of adulterants or compounds of different nature, type of production, etc., in addition to acting in the recognition and quantitative determination of concentrations of multiple components [2,9]. The performance of these systems can be improved with the use of electrodes or recognition layers with chemically manipulated surfaces, the insertion of sensitive modifiers/catalysts, such as carbonaceous materials (carbon nanotubes, carbon microspheres, graphene, etc.), conductive polymers or molecular imprint polymers, nanoparticles (NPs), nanocomposites, metallic phthalocyanines and enzymes [4].

In this context, carbon nanotubes (CNTs), since their initial report by Iijima in 1991 [10], have been indicated for a wide range of applications due to their one-dimensional structure (1D), which gives them excellent mechanical, thermal and electrical properties, in addition to low density and high aspect ratio length/diameter that give rise to their exceptionally large unit surface. These combined characteristics predetermine the electrochemical and adsorption properties of CNTs, which are responsible for their high responses and make them promising materials for the development of sensors [11,12]. Several reports show the application of CNTs in sensors (both single-walled carbon nanotubes—SWCNTs and multiwalled carbon nanotubes—MWCNTs, in different functionalizations) to detect different types of gas molecules, e.g., NH_3_, NO_2_, H_2_, CH_4_, CO and H_2_S [13,14,15], and volatile organic compounds (VOCs) such as benzene, toluene, methanol, ethanol, acetone, chloroform and tetrahydrofuran [16,17,18,19], which, when adsorbed on the surface of CNTs, change their conductivity, capacitance, dielectric constant, etc. [20].

To explore the unique characteristics of CNTs, much attention has been given to the macroscopic assembly of this nanomaterial into films called buckypapers (BPs) [21]. BP is a micrometer-thick planar thin film based on an aggregate of CNTs (or other carbon-based nanomateriais or nanocomposites) formed from van der Waals forces, exhibiting a porous entangled lattice structure [22,23]. These films, commonly produced by means of vacuum filtration of a well-dispersed solution of CNTs randomly distributed through a membrane filter, are flexible, easy to handle, low-cost, high-sensitivity and fast-response [22,23,24], and are promising for applications in a variety of motion sensors [25], temperature, pressure, humidity [24], biological and pharmaceutical compounds [21].

Shobin and Manivannan [26] reported the production of a chemoresistive sensor for the detection of concentrations between 62.5 and 1000 ppm of NH_3_ vapor at room temperature based on a BP of SWCNTs, using cellulosic filter paper as the substrate. However, the use of BP films for analysis in liquid solutions is still modest. Giordano et al. [20] reported the production of a self-sustaining BP of pure MWCNTs (obtained after removing the membrane support at the end of drying) applied to discriminate different concentrations of isopropanol in aqueous solutions and to analyze common beverages with increasing alcohol content, whose results showed that the film presents a reversible response with the change in electrical resistance proportional to the alcoholic strength of the solutions.

Mugo et al. [27] developed a flexible chemoresistive sensor based on a multilayer thin film with a fibrous nanoporous structure, produced by layer-by-layer deposition of nanocomposite cellulose nanocrystals (CNCs) and MWCNTs functionalized with carboxylic acid (COOH), incorporated in polyaniline (PANI) and dispersed on a 1 cm^2^ sheet of polyvinyl acetate (PVA). The PANI@CNC/CNT sensor exposed to the vapor of red wine samples (heated to 60 °C for 10 min) adulterated with methanol (0–7 vol.%) showed a change in electrical resistance with linear increase as a function of the concentration of methanol added. Abbeg et al. [28] reported the production of an integrated, smartphone-compatible handheld device (wi-fi and Bluetooth) formed by a highly sensitive chemoresistive sensor of palladium (Pd)-doped tin oxide (SnO_2_) NPs for rapid and selective detection of methanol (0.01–10 vol.%) and ethanol content in alcoholic beverages (beer, sake, wine, Baileys, arrack and Stroh rum) that were pure and adulterated with methanol.

In the present study, a BP film based on MWCNTs-COOH impregnated in cellulose fibers was used as a sensor for the analysis of Port wine, which was adulterated with percentages of 5 vol.% and 10 vol.% of distilled water and ethyl alcohol. The response of the sensor to ethyl alcohol, distilled water, and unadulterated Port wine was investigated to assess its ability to discriminate between adulterating agents added to wine and individual analytes. For this, the data were treated by multivariate statistical analysis using the PCA technique to estimate the feasibility of applying BP as a low-cost multisensor that is easy to operate and responds in real-time.

## 2. Materials and Methods

BP was produced by the vacuum filtration method using the same parameters applied in a previous study [29]. BP production steps are schematically illustrated in Figure 1. Initially, (1) MWCNTs-COOH supplied by the Federal University of Acre (with 99.80% purity) were dispersed in isopropyl alcohol PA ACS (60.1 g/mol and 99.50% purity, purchased from ÊXODO CIENTÍFICA, Sumaré, Brazil) using a model L100 ultrasonic washer (SCHUSTER, Santa Maria, Brazil) at 40 kHz for 60 min at room temperature, forming a 1:1 (g/L) solution of concentration that was (2) deposited on qualitative filter paper (weight: 80 g/m^2^, nominal thickness: 205 µm and pore size: 14 µm, J PROLAB, São José dos Pinhais, Brazil) by filtration through a Büchner funnel (12 cm in diameter, CHIARROT, Mauá, Brazil), fixed in 500 mL Kitasato (SPLABOR, Presidente Prudente, Brazil) under a nominal vacuum of 2.66 Pa (provided by a vacuum pump from SYMBOL TECNOLOGIA DE VÁCUO LTDA, Sumaré, Brazil). The formed film was (3) placed in an oven model SX1.0DTMS (STERILIFER, São Caetano do Sul, Brazil) at 100 °C for 1 h for the complete removal of humidity and solvent.

The morphological characterization of MWCNTs-COOH (as received) and MWCNTs-COOH/cellulose BP (as produced) was performed by scanning electron microscopy (SEM) using a microscope model VEGA3 (TESCAN, Brno, Czech Republic) at 20 kV. SEM micrographs were obtained via secondary electrons with working distances of 5.00 mm and 15.02 mm.

For the fabrication of the sensor, samples of cellulose-MWCNTs BP with an active area of 1.0 cm^2^ were cut and fixed on bakelite substrates and connected to copper electrodes (set apart by 1.5 cm) with the aid of pure silver conductive paint, as shown in Figure 2a. The electrical characterization of the sensor was carried out in the presence of distilled water, ethyl alcohol (Absolute 99.8% PA from Êxodo Científica, São Paulo, Brazil), Tawny Port wine with 19% ethyl alcohol content (purchased from Gran Porto Cruz, Vila Nova de Gaia, Portugal) and 19 vol.% distilled water/ethyl alcohol mixture (for comparison with the ethyl alcohol content of Port wine, as performed by Sysoev et al. in [30] for analysis of the influence of alcohol on the signal from the metal oxide nanobelt sensor they developed) measuring the electrical resistance as a function of time (R × t) by the two-point method using a digital multimeter model ET-2232 (MINIPA, Joinville, Brazil), where data collection was performed via device software. The analytes were added to the surface of the sensor using a process similar to that described by Yakhno et al. [31] with the aid of a single channel micropipette model Labmate (variable volume 0.1–10 µL, HTL SA, Warsaw, Poland) at room temperature, as shown in the schematic arrangement illustrated in Figure 2b, using intervals of 12 h between each analysis (interval required for natural evaporation of the analytes and drying of the sensor). Measurements were also performed on 10 mL samples of Port wine intentionally adulterated by the addition of 0.5 mL and 1.0 mL of distilled water and ethyl alcohol, composing a mixture of 5 vol.% and 10 vol.%, respectively.

To measure the electrical resistance of the samples over time, a digital picoammeter model 6487 (KEITHLEY, Cleveland, OH, USA) was used with the two-point method, since it allows measurements of electric currents in the range of 20 mA–2 nA. For comparison, these tests were performed on cellulose samples without MWCNTs-COOH (paper filter as received) in the same dimensions as the cellulose samples with MWCNTs-COOH (buckypaper as produced, Figure 2a) with and without the presence of Port wine. The pH of the analytes was measured using a digital benchtop pH meter model PHS-3E-BI (SATRA, Kettering, UK) with automatic calibration. The tests were performed on 10 mL samples, and at the end of each measurement the equipment probe was washed with distilled water to completely remove residues.

The discrimination capability of the sensor was investigated using the PCA technique, which is a powerful tool for pattern recognition, in which a set of multivariate input data is transformed into a coordinate space (two-dimensional, 2D, or three-dimensional, 3D) consisting of a smaller number of variables called principal components (PCs). PCA represents an unsupervised linear exploratory technique that provides a visualization of data variability, and is used to verify the interrelationships between objects, the relationship between objects and variables and the global correlation of the variables, detecting and interpreting patterns based on clusters of similarities and differences [4,19]. The total number of PCs generated is defined as a function of the number of input variables analyzed so the data set representation is delimited by the components that explain a percentage of total variance greater than 80% [32,33].

In this study, the data obtained from the analysis of response, response time and recovery time (reported as the main performance parameters of a sensor [34]) were used as input variables and treated by PCA using the free statistical software PAST v. 4.09 [35]. In the analysis, three weight columns were generated in the PCA, but only two were chosen because they explained a total variance of the data above 85.6%. However, to reduce the influence of the variables with greater weight on the generated PCs, the data were submitted to pre-processing using the relationship given in Equation (1):(1)$$z=\left(X-\overset{—}{X}\right)/s$$
where z is the z-score resulting from the linear transformation, X is the raw score measured for each sample, $\overset{—}{X}$ is the arithmetic mean of the raw scores, and s is the standard deviation of the sample distribution. The new autoscaled data present weight reduction and adjustment of the numerical difference of the variables.

## 3. Results

### 3.1. Morphological Characterization

The SEM micrograph in Figure 3a shows the MWCNTs-COOH received as flakes (inset), where it is possible to observe that the MWCNTs form an agglomerated morphology. The micrograph of Figure 3b shows that the morphology of the produced BP is based on a nanostructured sheet composed of a layer of MWCNTs-COOH (see red arrow in Figure 3b,c) distributed on the surface and between the micrometric cellulose fibers of the filter paper (see white arrow in Figure 3b,c), as evidenced in the high-magnification SEM micrograph in Figure 3c.

### 3.2. Electrical Response

The response of the sensor was calculated from the change in relative resistance normalized according to the relationship specified in Equation (2) [36], where $R_{f}$ is the final electrical resistance measured in the presence of analytes and $R_{0}$ is the initial electrical resistance of the sensor without analyte, both measured at room temperature.
(2)$$\mathrm{Response}=\left(R_{f}-R_{0}\right)/R_{0}$$

Sensor responses in the ON state for volumes between 0.1–6.0 µL of ethyl alcohol (pH 5.97) are shown in Appendix A, evidencing that the response (0.12 ± 0.03) was higher for 4 µL of solvent (see Appendix A). Compared in Figure 4, this result was lower than that obtained for the same volume of distilled water (pH 5.22), unadulterated Port wine (pH 3.26) and 19 vol.% water/alcohol mixture (pH 5.10), whose responses were 0.31 ± 0.05, 0.48 ± 0.10 and 0.20 ± 0.02, respectively, demonstrating that the response of BP in the presence of Port wine compared to ethyl alcohol was 3× higher. Furthermore, the dynamic response curves shown in Appendix A and in Figure 4 demonstrate periodic behavior of the increase (decrease) in the normalized electrical resistance of the sensor in the presence of analytes (after evaporation). Such behavior indicates a reversible and reproducible response after different analysis cycles.

Comparative measurements for a response cycle of cellulose samples without MWCNTs-COOH (filter paper as received) and cellulose with MWCNTs-COOH (buckypaper as produced) before and after exposure to 4 µL of unadulterated Port wine are presented in Appendix A. They demonstrate that MWCNTs-COOH, when exposed to the analyte, promote an increase in the normalized electrical resistance of the sensor. The different behavior observed for the cellulose sample without MWCNTs-COOH evidenced a decrease in its electrical resistance in the presence of the solution.

#### Port Wine Adulteration Analysis

The responses of the sensor to the presence of 4.0 µL of adulterated Port wine with distilled water are shown in Figure 5a. For adulterations with 5 vol.% and 10 vol.%, responses of 0.50 ± 0.04 (pH 3.70) and 0.75 ± 0.16 (pH 3.59), respectively, are observed, indicating an increase in the normalized relative resistance of the BP during the adsorption of these mixtures.

The responses for 4.0 µL of adulterated Port wine with ethyl alcohol are shown in Figure 5b. For adulterations with 5 vol.% and 10 vol.%, responses of −0.65 ± 0.10 (pH 3.90) and −0.21 ± 0.26 (pH 3.78), respectively, are observed, indicating a reduction in the normalized relative resistance of BP during the adsorption of these mixtures. Thus, it is noticeable that the response for ethyl alcohol presents a negative signal while the response for distilled water presents a positive variation. This fact shows that BP is able to detect the resistance variation in Port wine as well as discriminate between the two processes.

The response time (τ_Res_) and recovery time (τ_Rec_), collectively referred to as the “reaction time” of a sensor [34], are defined as the times required for the sensor to reach 90% of the total signal response in the presence of an analyte or target gas and return to 90% of the original baseline signal after removal of the sames, respectively [37]. In this study, τ_Res_ and τ_Rec_ were determined from the dynamic response curves, as shown in Figure 6a.

The comparative result of the mean response–recovery times for 4.0 µL of distilled water, ethyl alcohol, unadulterated and adulterated Port wine, shown in Figure 6b and summarized in Appendix A, shows τ_Res_ and τ_Rec_ of 78.0 ± 12.72 s and 181.5 ± 9.19 s for distilled water, 65.5 ± 5.74 s and 126.5 ± 12.23 s for ethyl alcohol and 95.5 ± 31.29 s and 163.0 ± 61.11 s for unadulterated Port wine, respectively. Adulteration of wine with distilled water caused an increase in the τ_Res_ and τ_Rec_ of the sensor, being 2240 ± 763.67 s and 2688 ± 876.81 s for 5 vol.% of water and 103.3 ± 70.94 s and 3035 ± 404.75 s for 10 vol.% water, respectively. The τ_Res_ and τ_Rec_ for wine adulterated with 10 vol.% ethyl alcohol increased to 105.0 ± 95.0 s and 870.5 ± 377.0 s, respectively. However, τ_Res_ for wine adulterated with 5 vol.% ethyl alcohol decreased to 10.0 ± 3.60 s, although τ_Rec_ increased to 1030.6 ± 116.86 s. At both concentrations, the τ_Rec_ for adulteration with alcohol was lower than that for water.

### 3.3. Multivariate Statistical Analysis

In the data processing using PCA, three PCs were generated (due to three input variables under analysis); however, only the first two were used (PC1 = 62.0% and PC2 = 23.6%), because together they explained a cumulative percentage of 85.6% of the total variance of the data. Appendix A shows the PCA Biplot generated, where the PCs were extracted via a covariance matrix (whose data are presented in Appendix A) in association with the vectors that correspond to the influence of the variables (response, response time and recovery time) on each PC.

Figure 7 shows a 3D diagram of sensor exposure to analytes as a function of the variables analyzed, which, similar to the PCA, shows an indication of the BP’s ability to discriminate between unadulterated and adulterated Port wine with 5 vol.% and 10 vol.% distilled water and ethyl alcohol, presenting the separation of scores of the samples in the graph. In relation to adulterated Port wine with distilled water, the PCA analysis also demonstrates an indication of the sensor’s potential to distinguish the percentage of distilled water added in the adulteration. Score data as a function of PCs for each analyte are shown in Appendix A.

## 4. Discussion

CNTs, due to strong intertubular van der Waals forces, tend to form bundles or cords, which results in high hydrophobicity, making their solubility in aqueous suspensions and in most common solvents unfeasible. Thus, the covalent bonding of functional groups such as COOH on the side walls/ends of CNTs as reported by Clark and Krishnamoorti [38] and Ma et al. [39] allows the improvement of their solubility/dispersion, providing continuous steric inhibition and avoiding the reformulation of the beams due to the formation of a stabilized colloidal system as a function of the electrostatic repulsion generated between the nanotubes and the dispersing medium. This repulsion is caused by the addition of surface charge on the CNTs because of the ionization of the functional group when interacting with solvents.

For this reason, in this study, the dispersion of MWCNTs-COOH in polar solvent occurred without the need for the addition of surfactants, whose preparation method and processing parameters used allowed us to obtain a BP film with a total thickness of 174.94 µm, formed by a layer of MWCNTs-COOH of approximately 67.33 µm uniformly dispersed over and between the cellulosic fibers (permeating approximately 40% of the filter paper) [29] as a result of the capillary forces suffered by the nanotubes through the pores during vacuum filtration. Furthermore, as reported by Shobin and Manivannan [26], the ease of absorption and evaporation of isopropyl alcohol by hydrophilic filter paper during filtration and drying, respectively, corroborates the effective binding of CNTs with paper.

Unlike the BP produced by Shobin and Manivannan [26], in which the cellulose fibers of the paper only acted as a substrate for the deposition of SWCNTs, in this study, they acted as a framework support for the layer of MWCNTs-COOH that adhered to its surface and impregnated its pores due to polar interactions between the hydroxyls of the functional group and cellulose, thus modifying the morphological characteristics of the paper and altering its nature as an electrical insulator. In the literature, Patole et al. [40] used a process of preparing BP by filtering graphene/pure MWCNTs through filter paper, but in the end, the paper was easily removed due to the weak surface interaction between the pure nanotubes and the cellulose polymer chains, resulting in a self-sustaining film, similar to that achieved by Giordano et al. [20].

The results presented in Figure 4 and Figure 5 show the BP detection capability for distilled water, ethyl alcohol, 19 vol.% water/alcohol mixture and unadulterated and adulterated Port wine with a response as a function of normalized electrical resistance variation, characterizing it as a chemoresistive sensor whose working principle is based on changes in electrical properties induced by mechanisms such as charge transfer, changes in electron scattering, contact effects and capacitance changes between analytes and CNTs in the chemical interaction process [41,42]. Thus, the analytes when added to the BP surface are adsorbed by the porous network of MWCNTs-COOH/cellulose fibers. This adsorption generates a change in the electrical conductivity of the nanotubes as a function of the polar interactions (dipole–dipole, hydrogen bonding) between the COOH functional groups and the analyte molecules, as indicated in the works of Kar and Choudhury [42] and Slobodian et al. [43], resulting in changes in electron scattering and, consequently, in the electrical resistance of the film, as confirmed by the results presented in Appendix A.

The response curves of the sensor demonstrate its reversible and reproducible character, in addition to pointing to the achievement of a distinct selective chemoresistive signal for each analyte, confirming, e.g., that the signal obtained for unadulterated Port wine containing 19 vol.% ethyl alcohol differs from the response obtained for ethyl alcohol diluted in distilled water at the same concentration (19 vol.%). In the presence of ethyl alcohol, as indicated in Appendix A, the response of BP tends to increase as a function of the increase in the volume of ethyl alcohol added, as reported by Giordano et al. [20] for the addition of volumes of 100 to 300 µL of isopropanol over a self-sustaining BP-based sensor. The sensor developed here had a pronounced response to the 4.0 µL volume of the solvent, which may be associated with the large chemical interaction with the CNTs/functional groups in this concentration range, resulting in a greater change in the conductivity of the nanomaterial.

The behavior of increasing/decreasing the relative resistance of BP for adulterated Port wine with distilled water and ethyl alcohol, respectively, may be a direct consequence of the intermolecular interaction of the adulterating agents added to the wine. In this sense, obtaining a maximum τ_Res_ of 2240 ± 763.67 s (approximately 37.3 min) and a maximum τ_Rec_ of 3035 ± 404.75 s (approximately 50.6 min) for wine adulterated with distilled water indicates the entrapment of water molecules in the MWCNTs-COOH network, culminating in an increase in reaction time, which promotes a more significant change in the electrical resistance of the sensor via electron trapping, thus causing a greater response compared to that achieved for individual analytes and adulterated wine with ethyl alcohol.

On the other hand, the process of interaction of the sensor with wine adulterated with ethyl alcohol predicts that the added alcohol molecules did not undergo entrapment, which corroborates the rapid interaction with the network of CNTs (smallest τ_Res_ of 10.0 ± 3.60 s), culminating in greater electron scattering and a decrease in normalized relative resistance, presenting a maximum τ_Rec_ of 1030.6 ± 116.86 s (approximately 17.2 min), much lower than the maximum obtained for adulterated wine with distilled water. These values, as well as the maximum τ_Res_ (95.5 ± 31.29 s) and τ_Rec_ (181.5 ± 9.19 s) obtained for the individual analytes, are lower than those reported for the sensor developed by Giordano et al. [20] with τ_Res_ of approximately 3600 s and τ_Rec_ of approximately 6–8 h, which may be a direct consequence of its morphology, with a response associated with the interaction of isopropanol/water solutions with the tangled networks of self-sustaining MWCNTs that were not subjected to functionalization processes by the adsorption of functional groups such as COOH, and do not use cellulosic fibers as a support structure for the formation of a layer of impregnated CNTs that permeate them.

The PCA Biplot generated in the multivariate analysis (Appendix A), as well as the 3D diagram in Figure 7, show an indication of the sensor’s distinguishability between unadulterated and adulterated Port wine with water and alcohol. PCA, however, allows the analysis of how the sample data correlate with the variables and both with the PCs. These PCs do not correlate with each other and are ordered so that the first component (PC1) explains or contains the largest variance of the available data in one dimension (relevant information), followed by the second component (PC2), which contains the maximum remaining information projected in this second direction (orthogonal to direction of PC1) [3].

Thus, the variables (response, response time and recovery) presented in the PCA Biplot through loading vectors, with origin in the center of the coordinate system of the PCs, present greater proximity to the axis referring to PC1, indicating that both variables are strongly correlated with this PC (which explains the highest percentage of data variance, 62.0%). In addition, the corresponding score for adulteration with 5 vol.% of water, located in the PCA in proximity to the vectors referring to the response and recovery times, demonstrates that these variables are correlated and have a greater influence on their results, as shown in Figure 6b. The score referring to adulteration with 10 vol.% water demonstrates that the same is strongly correlated with the response variable, according to the results presented in Figure 5b.

## 5. Conclusions

A BP film based on MWCNTs-COOH impregnated between cellulosic fibers was used to detect adulterating agents, such as distilled water and ethyl alcohol, intentionally added to a Tawny Port wine at concentrations of 5 vol.% and 10 vol.%. The response of the sensor investigated through normalized relative resistance curves demonstrates a periodic behavior in the presence of the analytes and after their evaporation, characterizing BP as a chemoresistive sensor with a selective, reversible dynamic response and reproducibility due to the modification of the electrical conductivity of the nanotubes caused by polar interactions between the functional groups COOH and the molecules of the analytes when they are adsorbed in the porous network of MWCNTs-COOH/cellulose.

The response of the sensor to adulterated Port wine shows a different behavior depending on the type of adulterating agent added, culminating in longer average reaction times for adulteration with distilled water due to the entrapment of molecules of the adulterant in the MWCNTs-COOH network, which causes the trapping of electrons and results in an increase in the variation of their normalized relative resistance. In relation to ethyl alcohol, the results point to a reduction in the BP response in the presence of adulteration with 5 vol.%, indicating that there was no trapping of molecules of the adulterant in the nanotube network due to the increase in the scattering of electrons, culminating, consequently, in the reduction of its normalized relative resistance and decrease of its reaction time.

PCA is a multivariate analysis tool that provides an indication of the sensor’s distinguishability by assigning scores to each sample and separating them in different quadrants on the graph. The data processing performed by the PCA associated with the analysis of the 3D diagram indicates the distinguishability of the sensor both to adulterated wine (with water or alcohol) and to the percentage of water added in the adulteration. The results allow us to infer the potential application of MWCNTs -COOH/cellulose BP as a low-cost, low-power-consumption, easy-preparation and easy-to-handle multisensor with real-time response in a future concept of nanostructured electronic tongue for adulteration detection in wine. However, for a complete assessment of the system’s ability to discriminate the analytes it would be necessary a greater number of measurements for each one. This process would result in clusters of points populated from similar samples, a factor that will be considered in future works.

## 6. Patents

Brazilian Patent application n° BR10202101028.

## Figures and Tables

**Figure 1 sensors-22-09732-f001:**
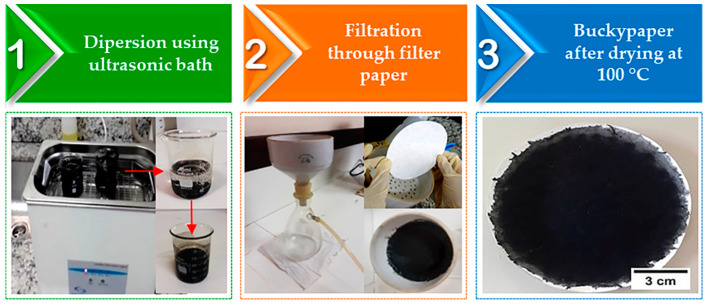
Schematic flowchart of BP production steps. (1) Dispersion; (2) filtration; (3) final buckypaper element.

**Figure 2 sensors-22-09732-f002:**
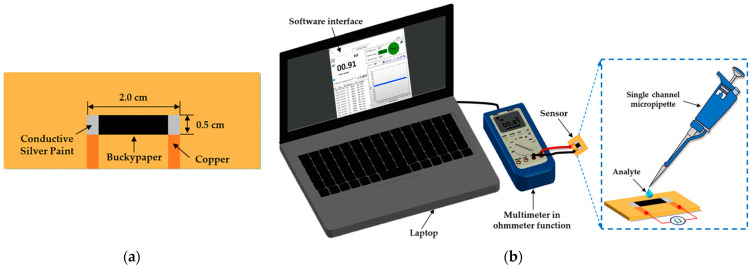
Schematic illustration showing (**a**) top view configuration and dimensions of the sensor and (**b**) the arrangement used in R × t measurements, where analytes were added with the aid of a single channel micropipette.

**Figure 3 sensors-22-09732-f003:**
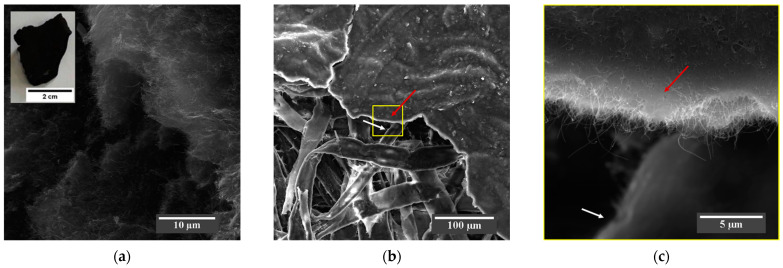
(**a**) SEM micrograph at 6600× magnification showing the agglomerated morphology of the MWCNTs-COOH flake (inset). The SEM micrograph of the BP in (**b**) at 518× magnification shows MWCNTs-COOH (red arrow) dispersed over and between the cellulose fibers of the paper (white arrow), which acts as a framework for the uniform deposition of the nanotubes, as shown in the SEM micrograph (**c**) at 11,000× high magnification in the region highlighted by the yellow square of (**b**).

**Figure 4 sensors-22-09732-f004:**
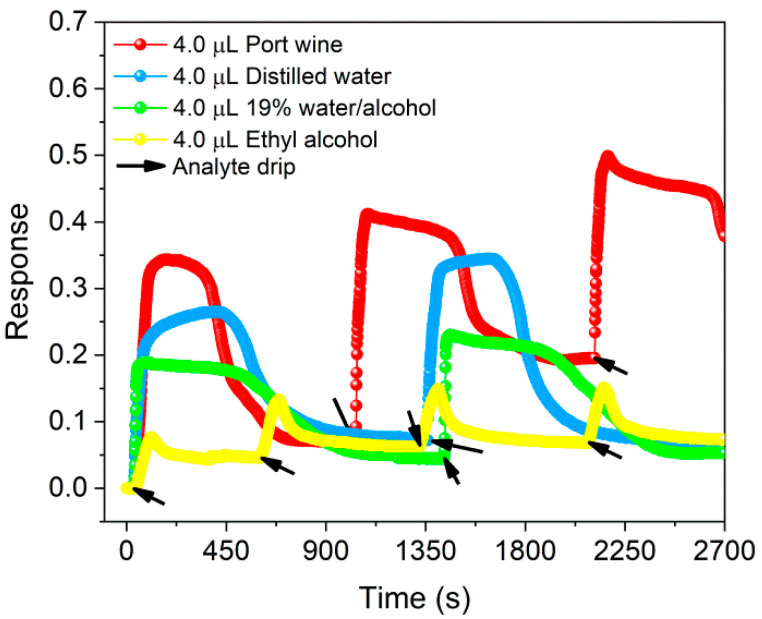
Response of BP in the presence of unadulterated Port wine, distilled water, ethyl alcohol and 19% water/alcohol mixture at volumes of 4.0 µL.

**Figure 5 sensors-22-09732-f005:**
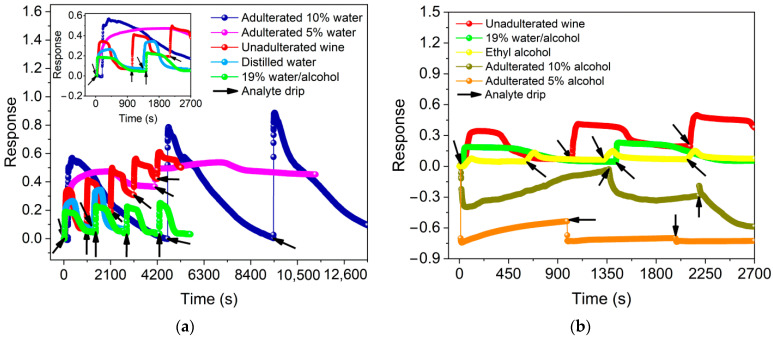
Sensor response in the presence of 4.0 µL of adulterated Port wine (**a**) with 5 vol.% and 10 vol.% distilled water compared to the same volume of distilled water, unadulterated wine and 19% water/alcohol mixture, and (**b**) with 5 vol.% and 10 vol.% ethyl alcohol compared to the same volume of ethyl alcohol, unadulterated wine and 19 vol.% water/alcohol mixture.

**Figure 6 sensors-22-09732-f006:**
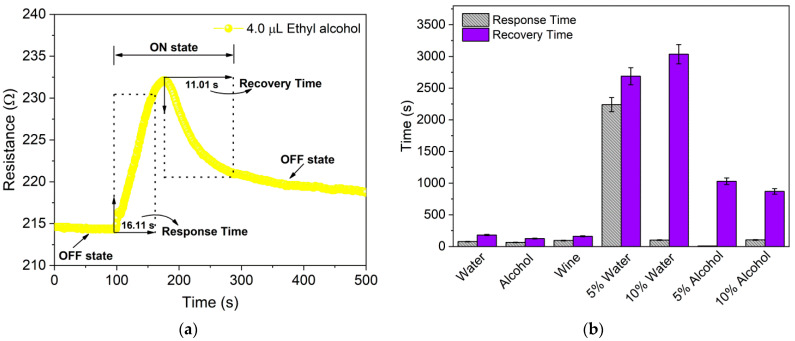
(**a**) Dynamic response curve employed to determine the response time (τ_Res_) and the recovery time (τ_Rec_) based on one cycle of the chemiresistor in the presence of 4.0 µL of ethyl alcohol. (**b**) Comparative mean response and recovery times for 4.0 µL of distilled water, ethyl alcohol, unadulterated and adulterated Port wine with 5 vol.% and 10 vol.% of distilled water and ethyl alcohol.

**Figure 7 sensors-22-09732-f007:**
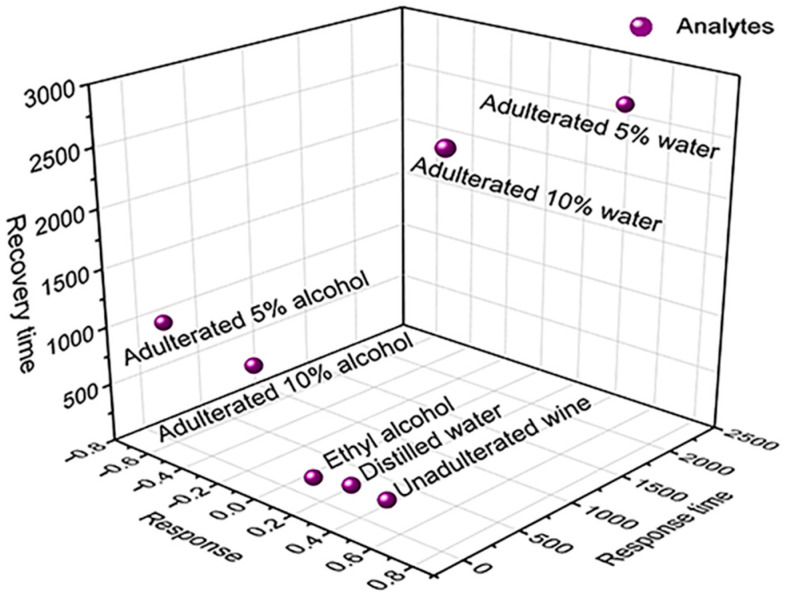
Three-dimensional diagram for 4 µL of ethyl alcohol, distilled water, unadulterated Port wine and adulterated with 5 vol.% and 10 vol.% of distilled water and alcohol as a function of response, response time and recovery.

## Data Availability

Not applicable.

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
