# Peer review of "Buckypaper-Based Nanostructured Sensor for Port Wine Analysis"

_sensors, 2022, doi:10.3390/s22249732_

Round 1
Reviewer 1 Report
The paper by L. Ferreira et al is about analyzing a wine’s quality via MWCNT/cellulose fiber sheets as chemiresistors. The author’s approach looks quite interesting and some similar to one reported in Sensors 20(2020) 5266, doi: 10.3390/s20185266.
While the data are possibly worth to publish in Sensors, the next issues should be resolved.
1. The Title of paper is about ”E-tongue”. However, the authors provide the data for a single sensor and, as it looks, process single sensor’s data (response and characteristic times as variables) for PCA analysis. So, to avoid reader’s confusing the authors are advised to change the title to something involving “… MWCNT sensor …” and remove “…E-tongue…”. Otherwise, the authors have to take the multisensor array and process its multidimensional vector signal.
2. While the authors discuss mostly MWCNTs as sensing element, it is unclear which material is responsible for the resistance’s change upon the analyte’s exposures. Therefore, the authors are advised to take the buckypaper without MWCNT as a blank reference and check its resistance under the analyte’s exposure to compare. This will be possibly a direct prove for MWCNTs as the sensing (adsorbing) element. Otherwise, the provided discussion around MWCNT has no a sense.
3. The authors employ distilled water, ethyl alcohol (ca. 99 %) and the wine as analyte probes with further wine’s dilution by 5 vol. % water and 10 vol.% of ethyl alcohol. Possibly, the noted contamination has a practice-origin idea but for the science purpose we need to properly compare the analytes in order to understand the nature of the signal. The data provided in Figs 4,5 should be supplemented with one more analyte’s exposure to equalize the effect of ethyl alcohol in probes of wine and alcohol (see for methodics: ACS Nano 4 (2010) 4487–4494, doi: 10.1021/nn100435h). With this purpose, the authors are advised to take the ethyl alcohol diluted with distilled water to get the same alcohol’s concentration as one in the wine (ca. 19 %) and compare the result with wine’s exposure. If authors could distinguish this alcohol/water mixture with the wine one, it would prove the method to be responsive to specific ingredients of the wine.
4. The authors employ PCA (Fig. 7) to distinguish the sensor’s parameters (response, response time and recovery time) to be characteristics of the analytes. With this purpose, they do not need PCA; they could just build 3D diagram where to plot the noted characteristics. Also, they have to measure these characteristics many times (not just once) and plot a set of data as a cluster.
5. The authors have to rename “sensitivity” throughout the text, introduced in Eq (2), to “response” according to accepted definitions; see IEEE Sensors J 1 (2001) 183-190, doi: 10.1109/JSEN.2001.954831.
6. The authors are advised to mark/highlight the analyte’s introduction in Figs 4,5. For further studies, they are advised to keep the exposure times equal to any analyte’s probes for a good research and proper presentation.
7. The authors are advised to check English grammar. See, for instance, p. 5, line 87: “… form randomly ordered agglomerated”.
Author Response
The paper by L. Ferreira et al is about analyzing a wine’s quality via MWCNT/cellulose fiber sheets as chemiresistors. The author’s approach looks quite interesting and similar to one reported in Sensors 20(2020) 5266, doi: 10.3390/s20185266.
While the data are possibly worth to publish in Sensors, the next issues should be resolved.
1. The Title of paper is about ”E-tongue”. However, the authors provide the data for a single sensor and, as it looks, process single sensor’s data (response and characteristic times as variables) for PCA analysis. So, to avoid reader’s confusing the authors are advised to change the title to something involving “… MWCNT sensor …” and remove “…E-tongue…”. Otherwise, the authors have to take the multisensor array and process its multidimensional vector signal.
The term “e-tongue” has been removed from the manuscript and title. The title has been changed to “Buckypaper-Based Nanostructured Sensor for Port Wine Analysis”. In the manuscript, the term “sensor element” was replaced by “sensor” only in p. 1, line 13; p. 3, lines 106,108,136,139; p.4, lines 146,166; p. 5, lines 205,208; p. 6, lines 209,218, 235; p. 8, lines 269,290; p. 9, lines 345,356,364; p. 10, lines 375, 378, 415; p. 11, lines 417,422,433 and by "multisensor" on p.1, line 22, p. 3, line 112; p. 11, line 436.
2. While the authors discuss mostly MWCNTs as sensing element, it is unclear which material is responsible for the resistance’s change upon the analyte’s exposures. Therefore, the authors are advised to take the buckypaper without MWCNT as a blank reference and check its resistance under the analyte’s exposure to compare. This will be possibly a direct prove for MWCNTs as the sensing (adsorbing) element. Otherwise, the provided discussion around MWCNT has no a sense.
Comparative analyzes between samples based on cellulose without MWCNTs-COOH (filter paper as received) and cellulose with MWCNTs-COOH (buckypaper as produced) with and without the presence of unadulterated Port wine were included in materials and methods on p. 4, lines 156 to 161 and the results are shown in Figure S2 of the supplementary material. Discussions of these results are presented on p. 6, lines 225 to 232; p. 9, lines 354,355.
3. The authors employ distilled water, ethyl alcohol (ca. 99 %) and the wine as analyte probes with further wine’s dilution by 5 vol. % water and 10 vol.% of ethyl alcohol. Possibly, the noted contamination has a practice-origin idea but for the science purpose we need to properly compare the analytes in order to understand the nature of the signal. The data provided in Figs 4,5 should be supplemented with one more analyte’s exposure to equalize the effect of ethyl alcohol in probes of wine and alcohol (see for methodics: ACS Nano 4 (2010) 4487–4494, doi: 10.1021/nn100435h). With this purpose, the authors are advised to take the ethyl alcohol diluted with distilled water to get the same alcohol’s concentration as one in the wine (ca. 19 %) and compare the result with wine’s exposure. If authors could distinguish this alcohol/water mixture with the wine one, it would prove the method to be responsive to specific ingredients of the wine.
The comparative analysis for ethyl alcohol diluted in distilled water at 19 vol.% was included in Figures 4 and 5. Discussions of these results are presented in p. 6, lines 213,214; p. 9, lines 358 to 360.
4. The authors employ PCA (Fig. 7) to distinguish the sensor’s parameters (response, response time and recovery time) to be characteristics of the analytes. With this purpose, they do not need PCA; they could just build 3D diagram where to plot the noted characteristics. Also, they have to measure these characteristics many times (not just once) and plot a set of data as a cluster.
The PCA plot was removed from the manuscript and transferred to the supplementary material (see Figure S3) and replaced with a 3D diagram (see Figure 7). In PCA data processing, 3 weight columns were generated (principal components), but only 2 were chosen because they provided more than 80% of score (data variance). This explanation was included in p. 4, lines 174 to 177; p. 5, lines 181,182 of materials and methods. Additional analyzes that can be obtained by PCA Biplot compared to the 3D diagram are emphasized in p. 8, lines 285 to 289 of the results and p. 10, lines 392 to 396 of the discussions.
5. The authors have to rename “sensitivity” throughout the text, introduced in Eq (2), to “response” according to accepted definitions; see IEEE Sensors J 1 (2001) 183-190, doi: 10.1109/JSEN.2001.954831.
The term “Sensitivity” has been replaced by “Response” throughout the text.
6. The authors are advised to mark/highlight the analyte’s introduction in Figs 4,5. For further studies, they are advised to keep the exposure times equal to any analyte’s probes for a good research and proper presentation.
Figures 4 and 5 have been altered and the point of introduction of analytes is exemplified by arrows in yellow. The curves were fitted to the same time interval (2,700 s). In figure 5a the inset shows the fitted region.
7. The authors are advised to check English grammar. See, for instance, p. 5, line 87: “… form randomly ordered agglomerated”.
The text was corrected appropriately, p. 5, lines 192,193.
Reviewer 2 Report
This paper presents the results achieved when using a buckypaper sensor to assess adulterated Porto wine.
There are some important aspects that need to be improved in this paper before it can be reconsidered for publication in Sensors. These are as follows:
1. The pH of the different solutions tested is not the same. Is pH the most important factor for the differences in sensor response observed? This could be elucidated by measuring different solutions with well controlled pH. Has this an impact in the discrimination capability of the sensor? In other words, should the falsified Port wine have the same pH of a real Port wine, would the system be able to find any difference between them?
2. The sensor is exposed to a liquid sample via drop-coating it onto the sensor surface. How is the sample 'removed' from the sensor? Is it left to evaporate? This needs to be clarified, as the 'recovery time' heavily depends on the methodology used.
3. The previous one is related to the fact that the reponse signals shown in Figures 4 and Figure 5 report different number of responses, recorded at different (random?) time intervals. This is weird.
4. The response and recovery times are similar for distilled water, ethyl alcohol and Port wine. However, the recovery times are strikingly different (much higher) for adulterated samples. Why it is so? Can authors give a plausible explanation for such a huge difference?
5. The sensor hardly recovers its original baseline value. Are sensors disposed of after mesuring (single use)?. How is the surface of the sensor is 'cleaned' from previous analytes? How many sensors were prepared and how reproducible they are?
6. The PCA plot shows a single data point for every type of sample. I think that the anbalysis should be reproduced for many replicate samples of each type, which would allow for checking the uncertainty of the measurements and how the resulting dispersion in data points affects the discrimination ability of the system.
7. Authors designate as 'sensitivity' what is in reality 'response'. Sensitivity is the change in sensor response caused by a change in analyte concentration.
Author Response
#REVIEWER 2
This paper presents the results achieved when using a buckypaper sensor to assess adulterated Porto wine.
There are some important aspects that need to be improved in this paper before it can be reconsidered for publication in Sensors. These are as follows:
- The pH of the different solutions tested is not the same. Is pH the most important factor for the differences in sensor response observed? This could be elucidated by measuring different solutions with well controlled pH. Has this an impact in the discrimination capability of the sensor? In other words, should the falsified Port wine have the same pH of a real Port wine, would the system be able to find any difference between them?
The pH of adulterated wine with 5 vol.% and 10 vol.% distilled water was included in the results on p. 6, line 237 and adulterated with 5 vol.% and 10% ethyl alcohol on p. 7, lines 246, 247. i.e., adulterated wine 5% water (pH 3.70), adulterated wine 10% water (pH 3.59), adulterated wine 5% alcohol (pH 3.90) and adulterated wine 10% alcohol (pH 3.78).
- The sensor is exposed to a liquid sample via drop-coating it onto the sensor surface. How is the sample 'removed' from the sensor? Is it left to evaporate? This needs to be clarified, as the 'recovery time' heavily depends on the methodology used.
Exactly, some samples like water and alcohol evaporate and promote different recovery times. A brief clarification of this question has been added in Materials and Methods, p. 4, lines 148,149.
- The previous one is related to the fact that the response signals shown in Figures 4 and Figure 5 report different number of responses, recorded at different (random?) time intervals. This is weird.
The time interval depended on the number of cycles that each analyte promoted as a function of its response/recovery time, for example, for wine adulterated with water, we had the greatest intervals. However, we have included an Inset showing the same time range for all analytes (see in Fig. 5a)
- The response and recovery times are similar for distilled water, ethyl alcohol and Port wine. However, the recovery times are strikingly different (much higher) for adulterated samples. Why it is so? Can authors give a plausible explanation for such a huge difference?
The increase in solubility of wine solutes by increasing the solvent is the main factor for the change in recovery time, in addition to the change in pH.
- The sensor hardly recovers its original baseline value. Are sensors disposed of after mesuring (single use)?. How is the surface of the sensor is 'cleaned' from previous analytes? How many sensors were prepared and how reproducible they are?
The sensor is single use and all that have the same electrical resistance are reproducible.
- The PCA plot shows a single data point for every type of sample. I think that the analysis should be reproduced for many replicate samples of each type, which would allow for checking the uncertainty of the measurements and how the resulting dispersion in data points affects the discrimination ability of the system.
Thanks for the suggestion. We are making new points to include in the next work.
- Authors designate as 'sensitivity' what is in reality 'response'. Sensitivity is the change in sensor response caused by a change in analyte concentration.
The term “sensitivity” was renamed to “response”. See p. 1, line 18; p. 2, line 66; p. 5, lines 178, 205; Equation 2; p. 6, lines 214,215,237; p. 7, line 246; p. 8, line 283; p. 9, line 361; p. 10, lines 365,376,409; p. 11, line 422; Figures 4, 5 and 7 of the manuscript and Figures S1, S2 and S3 of the supplementary material.

Round 2
Reviewer 1 Report
The authors have partially addressed the issues of concern. Still, the following questions remain.
1. According to comment #4, the authors still have to perform multiple repetitive measurements upon exposure to various analytes under study to plot the data in the Figs. 7 and S3 to prove the reliability of their claims.
2. According to comment #6, the Figs. 4,5 have still no “arrows” or something else to mark the sensor exposures to analytes.
3. The authors have not accounted for refs noted in the primary review.
4. English still requires heavy polishing.
Author Response
1. According to comment #4, the authors still have to perform multiple repetitive measurements upon exposure to various analytes under study to plot the data in the Figs. 7 and S3 to prove the reliability of their claims.
The affirmation of the sensor's distinguishability from the PCA biplot result was changed to just an “indicative of its discrimination ability” (see: p. 8, line 296 of the results; p. 10, line 392 of the discussions and p. 11, line 433 of the conclusions), as we used only a single point referring to each analyte. However, separating these into different quadrants in the PCA biplot allows estimating the system's ability to discriminate. A paragraph stating the need for a greater number of measurements for each analyte for a complete evaluation of the sensor's distinguishability was inserted in the supplementary material.
2. According to comment #6, the Figs. 4,5 have still no “arrows” or something else to mark the sensor exposures to analytes.
The color scale of the curves in Figures 4, 5 and 6 of the manuscript and S1 and S2 of the supplementary material were modified to highlight black arrows that signal the point of introduction of analytes on the sensor surface.
3. The authors have not accounted for refs noted in the primary review.
The indicated primary references were added to the manuscript. The reference ACS Nano 4 (2010) 4487–4494, doi: 10.1021/nn100435h has been included in Materials and Methods, reference [30], p. 4, line 150. The reference Sensors 20(2020) 5266, doi: 10.3390/s20185266 has been inserted into Materials and Methods, reference [31], p. 4, line 155. The reference IEEE Sensors J1(2001) 183-190, doi 10.1109/JSEN.2001.954831 was used in the definition of Equation 2, reference [36], p. 6, line 216.
4. English still requires heavy polishing.
The manuscript and supplementary information have been grammatically revised.
Please see the attachment.

Reviewer 2 Report
Authors have addressed most of my queries during their revision. The manuscript is now improved.
To my question nimber 6 "The PCA plot shows a single data point for every type of sample. I think that the analysis should be reproduced for many replicate samples of each type, which would allow for checking the uncertainty of the measurements and how the resulting dispersion in data points affects the discrimination ability of the system."
authors indicate that they will address this in the future. I accept that the paper can be publishable without performing new experiments. However, authors must include in their discussion of results a paragraph indicating that more measurements would be needed to study repeatability and completely assess the system's discrimination ability.
With this I consider the paper can be accepted for publication in Sensors.
Author Response
Authors have addressed most of my queries during their revision. The manuscript is now improved. To my question number 6 "The PCA plot shows a single data point for every type of sample. I think that the analysis should be reproduced for many replicate samples of each type, which would allow for checking the uncertainty of the measurements and
how the resulting dispersion in data points affects the discrimination ability of the system." authors indicate that they will address this in the future. I accept that the paper can be publishable without performing new experiments. However, authors must include in their discussion of results a paragraph indicating that more measurements would be needed to study repeatability and completely assess the system's discrimination ability.
This information was inserted in a paragraph in the supplementary material.
Please see the attachment.

Round 3
Reviewer 1 Report
The authors have improved the paper according to the comments. As a proof of concept, it's reasonable to leave single points characteristic for various analytes because the collecting more exposures would need obviously an extensive time. However, it will be imperative in practices to ensure the method works reliably. Therefore, the authors are advised to plan a number of repetitive measurements to examine in next research.
Author Response
The authors have improved the paper according to the comments. As a proof of concept, it's reasonable to leave single points characteristic for various analytes because the collecting more exposures would need obviously an extensive time. However, it will be imperative in practices to ensure the method works reliably. Therefore, the authors are advised to plan a number of repetitive measurements to examine in next research.
The authors are grateful for the reviewers' suggestions and guidance. For future work, repeat measurements and larger amounts of samples will be planned for system validation, as highlighted in p. 11, lines 427-430 of the manuscript.
